# Review of Elevated Para-Cresol in Autism and Possible Impact on Symptoms

**DOI:** 10.3390/ijms26041513

**Published:** 2025-02-11

**Authors:** Christina K. Flynn, James B. Adams, Rosa Krajmalnik-Brown, Alexander Khoruts, Michael J. Sadowsky, Khemlal Nirmalkar, Evelyn Takyi, Paul Whiteley

**Affiliations:** 1Biodesign Center for Health Through Microbiomes, Arizona State University, Tempe, AZ 85287, USA; ckflynn@asu.edu (C.K.F.);; 2School for Engineering of Matter, Transport, and Energy, Arizona State University, Tempe, AZ 85287, USA; 3School of Sustainable Engineering and the Built Environment, Arizona State University, Tempe, AZ 85281, USA; 4Department of Medicine, Division of Gastroenterology, Center for Immunology and BioTechnology Institute, University of Minnesota, Minneapolis, MN 55455, USA; 5Department of Medicine and BioTechnology Institute, University of Minnesota, Minneapolis, MN 55455, USA; 6ESPA Research, Sunderland SR5 3RL, UK

**Keywords:** p-cresol, autism, biomarker, microbiome, p-cresol sulfate, oxidative stress, mitochondrial dysfunction, kidney, metabolomics, dysbiosis

## Abstract

Para-cresol (p-cresol), and its primary human metabolite p-cresol sulfate (pCS), are among the most studied gut-derived metabolites relevant to autism spectrum disorder (ASD). P-cresol is produced by bacterial modification of phenylalanine or tyrosine and is one of many potentially deleterious metabolites produced by the gut microbiota. Seventeen studies have observed p-cresol and/or p-cresol sulfate as being higher in the urine of children with autism spectrum disorder (ASD) vs. controls. P-cresol has harmful effects on the body, including within the gut, brain, kidneys, liver, immune system, and mitochondria. Some of these effects may contribute to autism and comorbid symptoms. In the gut, p-cresol acts as an antibiotic, altering the gut microbiome to favor the bacteria that produce it. In the mitochondria, p-cresol disrupts ATP production and increases oxidative stress, which is also common in autism. In the brain, p-cresol impairs neuronal development. P-cresol inactivates dopamine beta-hydroxylase, which converts dopamine to noradrenaline. P-cresol sulfate impairs kidney function and is linked to chronic kidney disease (CKD), which is more common in ASD adults. P-cresol also interferes with immune function. Three animal studies have demonstrated that p-cresol causes autism-related symptoms in mice, and that mice can be recovered by the administration of fecal microbiota transplant from healthy mice. Similarly, it was found that microbiota transplant therapy treatment in children with ASD significantly reduced p-cresol sulfate levels to normal and led to significant improvements in gastrointestinal (GI) and ASD symptoms. In summary, p-cresol and pCS likely contribute to ASD core symptoms in a substantial subset of children with ASD.

## 1. Introduction

Autism spectrum disorder (ASD) is a complex neurodevelopmental condition defined by impairment in social interaction and social communication, repetitive behaviors, and restricted interests [1]. The prevalence of ASD is estimated at approximately 1 in 36 children aged eight years and presents with a 4:1 male to female gender bias [2]. Children diagnosed with ASD may exhibit a wide spectrum of cognitive, social, and behavioral manifestations [3]. The etiology of ASD and its associated symptoms is an active area of research, including multiple genetic and environmental factors.

Recent evidence suggests a link between the gut microbiota—the trillions of bacterial species that inhabit the gut—and ASD, highlighting the association between ASD and gastrointestinal dysfunction, as well as an altered microbial community structure and activity [4,5,6,7,8]. Approximately 40% of children with ASD have obvious functional gastrointestinal issues, such as diarrhea, constipation, bloating, and abdominal pain [9,10]. These symptoms may partially reflect abnormal gut microbiome composition and the production of potentially toxic metabolites, which could also contribute to ASD pathophysiology [8,11,12,13]. One such microbially derived metabolite is indoxyl sulfate, which has been found to be significantly elevated in six of six studies of ASD children vs. controls. It has multiple toxic effects on the body and brain and can cause ASD symptoms in animals [14]. Another important microbially derived metabolite is para-cresol (p-cresol) and its sulfated conjugate p-cresol sulfate (pCS). It is produced by gut bacteria via phenylalanine and tyrosine metabolism [15].

This review explores the role of p-cresol and pCS in ASD, beginning with their metabolic formation and subsequent processing within the human body. The studies of p-cresol and pCS in autism will be discussed based upon a systematic literature review, including 17 studies of urinary p-cresol and pCS concentration, which all found it to be higher in autistic children than typically developing children. This review will then examine the various toxicological classifications of p-cresol and pCS, including their functions as cytotoxins, neurotoxins, uremic toxins, and nephrotoxins. Additionally, this review will investigate other ASD comorbidities linked to elevated p-cresol and pCS levels. A discussion of a potential therapeutic approach (microbiota transplant therapy) for reducing elevated levels of p-cresol and pCS will be presented. Finally, recommendations for future research directions are provided.

### 1.1. P-Cresol Production

P-cresol is one of many potentially harmful metabolites produced by gut bacteria and has been observed at higher levels in autistic individuals [15,16]. P-cresol (see Figure 1 for structure) is produced by microbial metabolism of the aromatic amino acids tyrosine and phenylalanine in the colon [15]. There is no known human pathway capable of producing p-cresol [15,17,18]. Both phenylalanine and tyrosine are precursors of catecholamines, including dopamine, noradrenaline, and adrenaline, in humans, and p-cresol has been shown to directly interfere with metabolism and neurotransmitter synthesis [19]. As shown in Figure 1, the first metabolic step towards p-cresol generation is the conversion of phenylalanine to tyrosine mediated through enzymatic activity of phenylalanine hydroxylase. Then, through a series of metabolic reactions catalyzed by both humans (blue arrows) and bacteria (red arrows) [20,21].

At least 55 species of bacteria are known to produce p-cresol in the gut, but there may be other producers [15]. Saito et al. [15] reported that four gut bacterial species have been identified as primary p-cresol producers (*Blautia hydrogenotrophica*, *Clostridoides difficile*, *Olsenella ulli*, and *Romboutsia liuseburensis*), generating p-cresol 100 times more than the other 51 producers in their experiment. It is notable that *C. difficile* infections can cause severe life-threatening diarrhea [22]. P-cresol production is halted by the antimicrobial treatments neomycin and vancomycin [23], two antibiotics with minimal systemic absorption from the gut, demonstrating that bacteria producing p-cresol are susceptible to such pharmacotherapy.

#### Human Transformation of P-Cresol and Detoxification Pathways

Due to its unique chemistry, p-cresol disrupts multiple human metabolic pathways [24,25,26,27]. Humans process p-cresol through one of three known pathways that increase its excretion, as follows: (i) sulfation, (ii) glucuronidation, and/or (iii) binding to albumin [28]. The majority (between 88% and 95%) of total p-cresol and pCS in the blood is bound to albumin, while the unbound fraction is approximately 95% p-cresol sulfate and 5% p-cresol glucuronide in urine [29]. The amount of p-cresol in free form is negligible in urine due to its very low solubility in aqueous solutions [30]. When conjugated to albumin, the determination of p-cresol or pCS transport in the body becomes more difficult because albumin-bound substances in urine are rarely examined [28]. P-cresol and pCS are excreted in urine and feces [4,30].

## 2. Methods

A systematic literature search was conducted from 1 November 2022 to 2 January 2024 to identify papers that measured urinary p-cresol or pCS in patients with ASD vs. controls. Pubmed, Google scholar, and the Arizona State University (ASU) library website were searched using the following terms in various combinations: “ASD”, “Autism”, “Neurodevelopmental Delay” AND combinations of “Urinary” “Metabolomics” “Biomarkers” “metabolite(s)” “cresol” “p-cresol” “para-cresol” “4-cresol” “ρ- cresol” “p-cresol Sulfate” “p-cresyl sulfate” para-Cresol Sulfate” “4-Cresol Sulfate” “para-cresyl sulfate” “4-cresyl sulfate” “4-methylphenol”, and “p-methyl phenol”. Some studies reporting measurements of p-cresol or pCS in urine measured p-cresol or pCS sulfate with the ratio of pCS to p-cresol of 1000:1.

## 3. Results

### 3.1. P-Cresol Is Significantly Higher in Autistic Individuals than Controls Across the World

Urinary P-cresol or pCS is higher in individuals with ASD vs. controls around the world. Seventeen studies (see Table 1) from Italy, France, Georgia, Slovenia, Latvia, and China reported that p-cresol and its primary urinary metabolite pCS were elevated in children with ASD vs. typically developing (TD) controls [29,30,31,32,33,34,35,36,37,38,39,40,41,42,43,44,45]. The authors of 13 studies specified that this elevation reached statistical significance (*p* < 0.05). Three studies [33,43,45] also found elevations in urinary p-cresol in ASD individuals vs. controls but did not specify whether it reached statistical significance.

Although most of the studies compared urinary p-cresol concentrations or peak intensities against unrelated typically developing (TD) children, one study compared against sibling controls [34]. Those authors found that the mean relative concentrations of urinary p-cresol and/or pCS in children with ASD were double that of their unaffected (non-autistic) siblings.

Only five of the studies (Altieri et al. [29], Gabriele et al. [30], Tevzadze et al. [35], Li et al. [36], and Perisco et al. [37]) reported the mean values of p-cresol or pCS concentrations in their ASD groups and controls. All five reported similar qualitative results, and the concentrations of p-cresol or pCS were on the same order of magnitude for all studies. In those studies, mean p-cresol or pCS levels in ASD compared to controls ranged from 30% to 230% higher, with a median 90% higher concentration among the five studies. The quantitative values reported in each study varied, possibly due to differences in populations and/or study methods (See Figure 2).

Overall, these five studies demonstrate that a subset of children with ASD (19–26%) have abnormally high levels of urinary p-cresol or pCS.

### 3.2. Relationships of Age, ASD Severity, and Gender to P-Cresol and pCS Levels

Age Effects: Levels of urinary p-cresol or pCS may be higher in younger children. One study (Altieri 2011 [29]) found that urinary p-cresol was significantly higher in ASD in children under the age of 8 years (134 µg/mL for ASD vs. 70 µg/mL for TD, *p* = 0.005), but there was no significant difference in the urinary p-cresol concentrations in children aged 8–18 years (urinary p-cresol concentrations were 111.0 vs. 113, n.s.). However, our analysis of Osredkar et al. [38] found that there was no significant correlation with age for the ASD group or for the TD group, so the results of the two studies disagree and it is unclear if age affects levels of p-cresol in ASD.

ASD Severity Effects: Three studies investigated correlations of ASD symptom severity vs. level of p-cresol/pCS [29,30,39]. One study [29] found that p-cresol correlated with symptom severity of ASD for children below the age of 8 years, but not for those above 8 years of age. Furthermore, there were significant correlations (*p* < 0.05) of urinary p-cresol and/or pCS concentrations with several symptoms on the Childhood Autism Rating Scale, including imitation, use of the body, verbal communication, and general impression. Additional investigations were conducted on Altieri et al.’s [29] subset of ASD males, which found that urinary p-cresol concentrations were significantly positively correlated with the following three symptoms: imitation (R = 0.17), verbal communication (R = 0.58), and general impression criteria (R = 0.34) [29,30]. Similarly, another study by Mussap et al. [39] found that urinary p-cresol concentrations were 105% higher (*p* = 0.048) in children with severe ASD than those of children with mild-to-moderate ASD (ADOS-2 CSS ≤ 8). A third study by Osredkar et al. [38] also found urinary pCS levels correlated with the intensity of several ASD symptoms, including behavioral impairments and stereotypic and compulsive behaviors. They also reported that ASD children with additional diagnoses (epilepsy, hearing loss, delayed milestones, ADHD, and tic disorders) have significantly higher levels of p-cresol than ASD children without additional diagnoses. In summary all three studies concluded that total urinary p-cresol or pCS concentrations correlate with the severity of autism or several autism-related symptoms.

Gender Effects: Two studies also examined gender differences amongst their ASD subjects, namely Altieri et al. [29] and Osredkar et al. [29,38]. Altieri et al. [29] found that ASD females had higher urinary pCS concentrations than ASD males (188 µg/mL vs. 102 ug/mL, *p* < 0.05), whereas there was no significant difference in the control female group vs. males (71 ug/mL vs. 98 ug/mL., n.s.). Similarly, our analysis of Osredkar et al. 2023 [38] (see Figure 3) found that females had 2× higher average levels of pCS than males with ASD (averages of 173 vs. 88, *p* = 0.06), but the median values were 81 vs. 117, so a few females with unusually high values skewed the distribution. In contrast, the female controls had lower levels of pCS than the male controls (averages of 88 vs. 123, n.s.) Future studies with a larger proportion of females are needed to determine possible gender effects.

In summary, p-cresol and pCS are elevated in autistic individuals vs. controls in 17 studies and are possibly higher in younger children and females. Three studies [29,38,39] found that higher levels of p-cresol were associated with more severe ASD-related symptoms, and an additional study [30] found a correlation with repetitive and stereotypical behaviors but not overall severity. It is important to remember that correlation is not causation; however, p-cresol and pCS are harmful in multiple ways, so it is plausible that they contribute to some ASD-related symptoms and comorbid symptoms.

### 3.3. Health Affects of P-Cresol and/or pCS

#### P-Cresol May Be Linked to Catecholamine Abnormalities in ASD

Increases in urinary catecholamine levels have been linked to high levels of p-cresol [19]. This may be due to the inhibition of the enzyme dopamine beta hydroxylase by p-cresol, and by diverting the parent compound of both dopamine and p-cresol toward microbial metabolism rather than neurotransmitter synthesis. P-cresol inhibits dopamine beta hydroxylase, which converts dopamine to noradrenaline, which could result in excess dopamine in the brain [46]. Notably, decreased levels of noradrenaline and adrenaline have been noted in ASD individuals when compared with their typically developed peers (*p* < 0.001) [40]. Furthermore, Gevi et al. [40] discovered that dopamine levels correlated exponentially with urinary levels of *p*-cresol in ASD children whose urine contained high levels of p-cresol.

### 3.4. Possible Contribution of P-Cresol and pCS to Sulfation Abnormalities in ASD

Decreased sulfation capacity and decreased plasma sulfate have been reported in individuals with ASD [47,48,49,50]. P-cresol may be more toxic in children with autism because those children likely have a decreased ability to excrete it due to a greatly decreased sulfation capacity. Challenge tests with acetaminophen (paracetamol) are a standard way to determine sulfation capacity [51]. Specifically, comparing ASD participants with non-ASD participants ingesting an oral 250 mg dose of acetaminophen (paracetamol), the ratio of acetaminophen sulfate vs. acetaminophen glucuronide in the urine is 0.98 for ASD vs. 3.4 in non-ASD children (*p* < 0.00002) [49], demonstrating that sulfation capacity is greatly reduced in ASD. Other studies have yielded similar results [48,50].

The decrease in sulfation capacity in autism is likely due to the following factors: (1) decreased sulfate [47,52], (2) decreased activity of phenol sulfotransferase enzymes (PST) [53], and/or (3) increased competition for sulfation due to elevated levels of p-cresol and possibly other toxic substances that require sulfation. Thus, elevated levels of p-cresol place an additional burden on the limited sulfation ability of children with autism. Thus, p-cresol is likely more toxic to children with autism because they have a reduced ability to sulfate it for excretion.

### 3.5. pCS Was Elevated in Feces of ASD Individuals and Decreased After Microbiota Transplant Therapy (MTT)

pCS may be higher in the feces of children with ASD, as well as in their urine [7]. Kang et al. discovered 2.57-fold higher levels of pCS in fecal samples of ASD children with gastrointestinal disorders (constipation and/or diarrhea) when compared to typically developing children (*p* = 0.08) [54]. The ASD children were then treated with microbiota transplant therapy (MTT), which involved antibiotic treatment, then a bowel cleanse, and then 7–8 weeks of daily microbiota transplant from very healthy donors. MTT resulted in a substantial improvement in GI and ASD symptoms, and the fecal pCS levels decreased by 91% over the course of the 18-week trial, becoming similar to the levels in the TD group. Upon reanalysis of the same data using an updated database by Nirmalkar et al., p-cresol was one of only five metabolites that were significantly elevated in feces at baseline that significantly decreased at the end of treatment [8]. The observed change in fecal pCS levels after MTT provides additional evidence that increased levels of pCS are due to gut bacteria, and that a reduction in pCS levels was accompanied by substantial clinical improvement in an open-label study, as well as with microbiome improvements. Overall, this study supports the previous studies of urine that found increased p-cresol and/or pCS in children with ASD, and the large decrease in levels of pCS after microbiota transplant in parallel with improvements in GI and ASD symptoms suggests that pCS may contribute to both GI and ASD symptoms [9,55,56].

### 3.6. P-Cresol Toxicity to Various Organ Systems

ASD involves many core and co-morbid symptoms, including neurological impairment, gut problems, impaired mitochondrial function, oxidative stress, immune system dysregulation, and increased rates of chronic kidney disease and Parkinson’s disease. P-cresol and pCS have been shown to affect all of those symptoms, so it is likely that elevated p-cresol can also contribute to those symptoms in ASD [13,57,58,59,60,61,62]. Table 2 and Figure 4 below summarize the harmful effects of p-cresol and/or pCS by organ system.

#### 3.6.1. P-Cresol Causes Mitochondrial Dysfunction and Oxidative Stress

Many studies have demonstrated both mitochondrial dysfunction [63] and increased oxidative stress [5] in ASD, and p-cresol is likely a contributor to those problems [64]. P-cresol and pCS impair mitochondrial function, resulting in increased production of reactive oxygen species and causing increased oxidative stress, which damages kidneys [64]. In human kidney cell cultures, clinically relevant concentrations of p-cresol and pCS lead to a concentration-dependent rise in reactive oxygen species (ROS) and a significant decrease in cell viability [65]. Renal oxidative stress induced by pCS results in enhanced activity of NADPH oxidase 4 (NOX4) within mitochondria and elevated intracellular hydrogen peroxide levels. A downstream effect of NOX4 activity changes results in the inactivation of respiratory chain complex I, therefore, less ATP is produced.

Research has shown that pCS, along with other uremic toxins like indoxyl sulfate, negatively affect mitochondrial function [64,66]. Kozeiel et al. asserts that sustained NOX4 activity disrupts mitochondrial function by interfering with the electron transport chain complexes in endothelial cells, while Edamatsu et al. confirmed that ROS production increases with exposure to these toxins [66]. Additionally, Sun et al. observed mitochondrial damage and reduced mitochondrial mass in renal tubular cells exposed to p-cresol [67,68].

P-cresol and pCS also impair mitochondrial respiration linked to NAD and succinate, potentially inducing mitochondrial swelling. One study suggests that p-cresol’s and pCS’s hepatotoxic effects may stem from such swelling or increased phospholipase A2 activity, which disrupts fatty acid metabolism [69]. Furthermore, the in vitro studies reported by Yan indicated that certain Cytochrome P450 enzymes convert p-cresol into reactive quinone methides, which cause oxidative stress and require glutathione for detoxification. Inadequate glutathione levels can lead to the formation of harmful adducts with membrane-bound proteins [69].

Inhibition of mitochondrial function can inhibit cell proliferation. Li et al. [70] found that both free and protein-bound p-cresol can inhibit the proliferation of human umbilical vein endothelial cells by up to 80%, causing cell cycle arrest in a dose-dependent manner [70,71].

Overall, p-cresol and pCS disrupt mitochondrial function, leading to increased oxidative stress, reduced ATP production, and glutathione depletion [72]. Decreased mitochondrial function can adversely affect nearly all cells and organ systems in the body.

#### 3.6.2. Gastrointestinal Effects of P-Cresol

Approximately 40% of children with autism have chronic functional gastrointestinal disorders (constipation, diarrhea, bloating, and abdominal pain), and many also have accompanying abnormal gut bacteria and increased gut permeability [72]. Several studies strongly suggest that p-cresol likely contributes to these problems. P-cresol has multiple effects on the GI tract. P-cresol is reported to have antibiotic-like properties. Gram-negative bacteria like *Escherichia coli*, *Klebsiella oxytoca*, and *Proteus mirabillis*, and other gammaproteobacteria-class bacteria, were selectively inhibited by p-cresol in vitro [22,73]. P-cresol has been suggested to kill butyrate-producing microorganisms within the colon [74]. *Clostridia* and some other bacteria can withstand high p-cresol concentrations while also producing the molecule. This high tolerance to p-cresol provides a competitive advantage for p-cresol-producing bacteria [58]. Thus, once established, infections of p-cresol-producing bacteria like *Clostridioides difficile* may be self-perpetuating.

In addition to modifying the microbiome, p-cresol also has effects on intestinal physiology. P-cresol is genotoxic to colonocytes, leading to cancer-causing mutations. Increased levels of p-cresol are associated with colon cancer [58]. P-cresol also weakens tight junctions in the GI tract, resulting in increased intestinal permeability [58]. Rates of cancer, including colon cancer, are significantly higher in ASD individuals vs. controls (Odds Ratio (OR) 1.3 (*p* < 0.05) and 1.8 (n.s.), respectively), with the rates being higher only in those with intellectual disability or birth defects [75]. One possible explanation of this finding is that higher levels of p-cresol cause both more intellectual disability and a higher risk of cancer. In one autism study, Gabriele et al. [30] found that elevated levels of urinary p-cresol sulfate correlated with decreased motility and chronic constipation in children with ASD [76]. The authors further found that urinary p-cresol sulfate was correlated with chronic constipation in ASD.

In summary, p-cresol has multiple effects on the gut, including altering the gut microbiome via its antibiotic effect, weakening tight junctions, possibly contributing to colon cancer, and correlating with chronic constipation in ASD.

#### 3.6.3. Neuronal Toxicity of P-Cresol

P-cresol likely contributes to many of the neurological problems present in children with autism, including effects on ASD symptoms, neuronal development, neuronal function, oxidative stress, and neuroinflammation. In mice, p-cresol has been reported to cause ASD-related symptoms, neuroinflammation, and neuronal loss [77]. These authors found that daily oral gavage of p-cresol given to mice induced ASD-like social deficits and increased observed repetitive behaviors. A significant positive correlation of p-cresol with *Clostridiodes* (*p* < 0.001) and *Desulfovibro* (*p* < 0.01) was also found, and treating these animals with fecal microbiota transplant (FMT) from non-dysbiotic animals recovered the ASD-like behaviors and social deficits [77]. Liu et al. separately discovered that daily injection of p-cresol/DMSO solution for 14 days in four-week-old mice resulted in increased autistic-like symptoms, including increased locomotor activity, increased anxiety-like behaviors (*p* < 0.01), and increased social impairment (*p* < 0.001) compared to both naïve and saline-control animals [78]. P-cresol significantly (*p* < 0.01) altered the relative expression in various genes, including NMDA1, KEAP, and others, in the hippocampus [79]. Furthermore, animals treated with p-cresol experienced increased apoptosis of neuronal cells (due to cleavage of caspase 3) cleaved within the hippocampus compared to both naïve and vehicle-control animals. P-cresol treatment resulted in neuronal loss in the dentate gyrus and hippocampal region CA1 due to overexpression of apoptotic signals and enhancement of neuroinflammation [80]. Overall, these studies found that the administration of high levels of p-cresol can cause ASD symptoms in mice, which suggests that it may contribute to ASD symptoms in humans, although differences in species, dosage level, and duration of administration should be kept in mind [77,80].

pCS has multiple effects on neuronal development that may also contribute to ASD symptoms. pCS crosses the blood–brain barrier via membrane-bound organic acid transporters or via the loss of tight junctions [81]. Guzman-Salas et al. demonstrated that p-cresol impairs dendritic development in Na2 and PC-12 neuronal cell lines in a dose-dependent fashion [80]. A significant decrease in the number of primary and secondary dendrites, neurite length projections, and arborization shorter than 80 µm were also observed in a dose-dependent fashion. Other studies have yielded similar results [79,80]. Tevzadze et al. also discovered decreased neurite length, increased production of brain-derived neurotrophic factor (BDNF) proportional to increasing p-cresol and pCS concentrations, and that low doses of p-cresol increased the expression of NF subunits [61]. Additionally, brain-derived neurotrophic factor (BDNF) stimulates brain growth during early development and could be involved in brain hypergrowth shown in autistic children on MRI scans and upon postmortem analysis that occurs between ages 2 and 4 years [82]. NF interacts with GluN1 subunits of NMDA receptors and D1 Dopamine receptors, and abnormal NF levels impact neuronal cell structural remodeling [61]. Finally, it has been demonstrated that pCS, as well as other uremic toxins like ammonia and indoxyl sulfate, can accumulate within the brain and cerebral spinal fluid in those with Parkinson’s disease, causing oxidative stress and inflammation, which may contribute to neurological impairments and encephalopathy [38,81]. Thus, early exposure to p-cresol or pCS may significantly impact overall brain development.

In summary, p-cresol and pCS have multiple effects on neuronal development, neuronal function, oxidative stress, and neuroinflammation, can cause ASD symptoms in mice, and likely contribute to neurological problems and ASD symptoms in humans.

#### 3.6.4. Nephrotoxicity of P-Cresol Sulfate

Adults with autism have increased rates of chronic kidney disease, and elevated levels of pCS likely contribute to kidney problems. In uremic patients, p-cresol and pCS accumulate in the lungs, liver, blood, urine, and kidneys, with post-mortem levels between 20- to 40-fold higher in these tissues than those found in healthy controls [28]. Furthermore, high levels of pCS and indoxyl sulfate, another microbially derived toxic metabolite, are often prognostic indicators utilized to predict mortality outcomes

pCS uses the organic acid transporter (OAT3) to enter the renal tubular cells and to cross the blood–brain barrier [67], where pCS triggers free radical production. The resulting oxidative stress leads to enhanced cytokine expression and enhances nuclear factor kappa B expression in those with chronic kidney disease (CKD) [78]. The induction of oxidative stress and inflammatory response to pCS have been demonstrated to greatly interfere with filtration efficiency in CKD patients and decrease renal tubular cell function. Miot et al. discovered that 25.6% of adults with a mean age of 42 years with ASD had CKD [83].

#### 3.6.5. Hepatic Effects

Children with autism have increased levels of oxidative-stress-related compounds, and this is one of the contributing factors to elevated levels of p-cresol [84]. Cytochrome P450 enzymes in liver microsomes convert p-cresol into quinone methides, which are highly reactive oxygen species [69]. P-cresol can also affect the liver by competing for detoxification resources (sulfate, glucuronide, and albumin) with naturally occurring metabolites, as well as by placing an additional energy burden on enzymatic activity [24].

#### 3.6.6. Immunological Effects of P-Cresol

Many studies have reported alterations in immunological function in ASD [68], and p-cresol may contribute to some of those problems. P-cresol and its host-mediated metabolites have long been considered one of the main uremic toxins responsible for an immunodeficient state of patients with chronic renal failure (CRF) [85,86]. Specifically, p-cresol has been shown in vitro to depress phagocytic activity [87], inhibit cytokine production by macrophages [88,89], and decrease cytokine-induced endothelial adhesion molecule expression and leukocyte trafficking [89,90,91,92]. However, the contribution of p-cresol and pCS to systemic immunosuppression has been called into question because p-cresol is rapidly metabolized by the host [93]. Nevertheless, it is possible that p-cresol does interact directly with the mucosal immune system in the intestine.

Interestingly, despite its association with immunodeficiency, CRF is also characterized by a state of chronic, low-grade inflammation [57,94,95]. The two phenomena are not necessarily contradictory and can be both driven by chronic stimulation of microbe-associated molecular receptors resulting in a state of ‘immune paralysis’ or endotoxin tolerance [12]. CRF is associated with the weakening of the gut barrier function, at least in part secondary to the decreased expression of proteins forming the epithelial tight junctions and a reduction in transepithelial resistance [96]. Some of these effects can be driven by p-cresol, which has been demonstrated to have deleterious effects on intestinal epithelia [97].

Both pCS and p-cresyl glucuronide have been linked to endothelial dysfunction [98], altered endothelial–leukocyte crosstalk [99], and diminished leukocyte activation in CRF [26]. Interestingly, p-cresol sulfate suppresses the production of interferon-γ by Th1 cells and inhibits Th1-mediated contact hypersensitivity response in vivo [100]. This effect may be driven by p-cresyl sulfate interference with antigen processing [101]. Notably, the suppressive effects of p-cresyl sulfate on peripheral monocytes and macrophages have also been shown in brain microglial cells. The exposure of microglia to p-cresyl sulfate results in the attenuated release of TNF-α and IL-6 and decreased phagocytic activity [102]. These effects on microglia are associated with decreased levels of A Disintegrin and Metalloprotease 10 (ADAM10) and A Disintegrin and Metalloprotease 17 (ADAM17) proteins, which may also contribute to synaptic dysfunction and altered brain connectivity in ASD [103,104]. P-cresol and pCS can also trigger the increased expression of DNA Methyltransferase enzymes 1, 3a, and 3b in vitro, demonstrating that DNA methylation can be affected by these two compounds [78]. DNA methylation may lead to epigenetic upregulation of inflammatory genes via histone modifications.

In summary, p-cresol and pCS induce immune dysregulation via the depression of phagocytic activity of macrophages, a reduction in Th1 cytokine production, an alteration of leukocyte trafficking, and an alteration of gut barrier function.

### 3.7. Other Diseases with Links to High Levels of P-Cresol and/or P-CS

#### 3.7.1. Parkinson’s Disease

Adults with ASD have a much higher rate of Parkinson’s disease than the general population. Miot et al. [83] reported that 39.6% of adults with ASD had Parkinson’s disease, beginning at a mean age of 42 years. Another systematic review found that parkinsonism and Parkinson’s disease were much more common in ASD individuals than in the general population [105]. P-cresol likely contributes to the increased rate of Parkinson’s disease in autism because it has also been found in higher levels in individuals with Parkinson’s disease [106,107,108]. It has been reported that patients with Parkinson’s disease have a nine-fold increase in the ratio of cerebral spinal fluid to plasma pCS concentrations compared to healthy controls [81], suggesting a major build-up of pCS in the brain. They also noted increased Mono Amine Oxidase activity in Parkinson’s disease patients, which may involve altered tyrosine metabolism. People with Parkinson’s often develop GI disorders many years prior to neurological symptoms, suggesting that increased levels of toxic gut metabolites, such as pCS, can eventually result in neurological effects in adults [109,110,111]. P-cresol inhibits dopamine beta hydroxylase, which converts dopamine to nor-epinephrine, which would result in excess dopamine in the brain [46,112].

#### 3.7.2. Epilepsy

About 30% of people with ASD will develop epilepsy at some point in their lives, and up to 82% have abnormal epileptiform [113]. In most cases of epilepsy, no underlying causative condition is known, either in autism-related or in general epilepsy. Elevated p-cresol or pCS may contribute to a significant fraction of those cases by altering neurotransmitter function.

In rodent studies, treatment with p-cresol induced seizures by modifying the expression of two subunits (GLUN2A and GLUN2B) of N-Methyl-D-Aspartate receptors found within the hippocampus and Nucleus Acumens [114]. Also, p-cresol inhibits the conversion of dopamine to noradrenaline, which may affect seizure risk. In humans, one small study [35] investigated urinary p-cresol concentrations in children aged 1–4 years with ASD without epilepsy (n = 14), age-matched counterparts with epilepsy but without autism (n = 14), and age- and sex-matched typically developing children (n = 14). They discovered that those with epilepsy had the highest urinary p-cresol concentrations (mean p-cresol 119.5 ± 9.17 µg/mL), followed closely by those with ASD (mean p-cresol 102.6 ± 12.72 µg/mL), much higher than urinary concentrations of the typically developing children (mean p-cresol 31.5 ± 10.74, µg/mL).

### 3.8. Summary of Effects of P-Cresol and pCS on ASD Symptoms and the Body

As shown in Table 2 below, p-cresol and pCS have many adverse effects on the gut, mitochondria, brain, liver, kidneys, and immune system. The effects on mitochondria, including the increase in oxidative stress and decrease in production of ATP, affect nearly every cell and organ in the body and are an important part, but not the only part, of their toxic effects. The effects on many organ systems may contribute to many ASD symptoms and common comorbid ASD symptoms, including gut problems, neurological symptoms, mitochondrial dysfunction, oxidative stress, and an increased rate of chronic kidney disease and Parkinson’s disease. Elevated levels of p-cresol may also contribute to increased rates of epilepsy and abnormal epileptic activity in autism.

## 4. Discussion

### Why Is P-Cresol Elevated in Children with ASD?

There are multiple possible factors that may increase levels of p-cresol and/or pCS in autism. One small study [6] found that ASD children with gastrointestinal symptoms (constipation and/or diarrhea) compared to TD children had higher rates of C-section birth, reduced duration of breastfeeding, increased use of antibiotics during infancy, and reduced fiber consumption, all of which could contribute to abnormal gut bacteria and the overgrowth of p-cresol-producing bacteria. Also, decreased sulfation ability in ASD due to genetics [53] or decreased intake [115] results in a decreased ability to excrete p-cresol, likely resulting in higher levels of it in the gut, which enhances the growth of the bacteria that produce it by acting as an antibiotic against competing bacteria.

## 5. Future Research Directions

P-cresol and pCS are elevated in a significant subset of children with ASD, as well as people with chronic kidney disease, Parkinson’s disease, and possibly epilepsy. There are several unanswered questions that warrant future investigation regarding p-cresol and pCS and their impact. First, it would be helpful to determine which bacteria are the primary producers of p-cresol in different human disorders and if elevated levels of p-cresol and pCS are associated with specific alterations in the microbiome. It may be possible to identify a microbiome signature of high p-cresol production potential. Second, it would also be helpful to identify the major microbial competitors for bacteria that produce p-cresol, which could help in developing treatments for elevations in p-cresol. Third, it would be helpful to know what factors promote the growth of bacteria that produce p-cresol. Fourth, it would be useful to conduct further treatment studies to confirm if MTT and/or other treatments are effective in reducing p-cresol and the bacteria that produce it and to confirm if those reductions result in improved symptoms.

Since many bacterial species have some of the enzymes responsible for the conversion of tyrosine and phenylalanine to p-cresol, much can be learned from analyzing the intestinal bacterial metagenome of ASD individuals for key enzymes, including p-HPA decarboxylase, tyrosine aminotransferase B, and tyrosine lyase. Coupled with metabolomic analysis of feces and urine, p-cresol levels may provide insight into the gut metabolism leading to the production of harmful metabolites. In addition, future studies should also use transcriptomic analyses to gain a better understanding of the coordinated expression of enzymes responsible for the synthesis and catabolism of p-cresol sulfate by gut. This will ultimately provide the necessary metabolic insight needed for interventional studies on ASD individuals receiving MTT and dietary changes to result in lessening ASD symptoms.

## 6. Conclusions

P-cresol and pCS are harmful gut gastrointestinal metabolites that have been found to be ~35–230% higher in the urine of children with ASD vs. non-autistic control participants. Three studies found that levels of p-cresol or pCS significantly correlated with some measures of ASD-related symptom severity [29,30,39].

P-cresol has been found in three studies to cause ASD-like symptoms when administered to animals [19,77,114]. P-cresol and pCS also likely contribute to many core and co-morbid ASD symptoms in humans, including neurological problems, gut problems, mitochondrial dysfunction, oxidative stress immune system dysregulation, and increased rates of chronic kidney disease and Parkinson’s disease. They may also contribute to increased rates of epilepsy and abnormal epileptiform activity in ASD.

Levels of pCS in feces of children with ASD were elevated compared to those of controls and greatly decreased after microbiota transplant therapy, accompanied by significant improvements in GI and ASD symptoms [8]. This study was consistent with a study of mice who were administered p-cresol, which caused ASD symptoms, and then were recovered by treatment with fecal transplants from healthy mice.

P-cresol is just one example of harmful bacterial metabolites in the gut that may contribute to autism symptoms. Future work is needed to investigate other microbial-produced metabolites that may also contribute to ASD symptoms and co-morbid symptoms. Gastrointestinal infections that produce p-cresol and other microbial metabolites may affect neurological function in autism. 

## Figures and Tables

**Figure 1 ijms-26-01513-f001:**
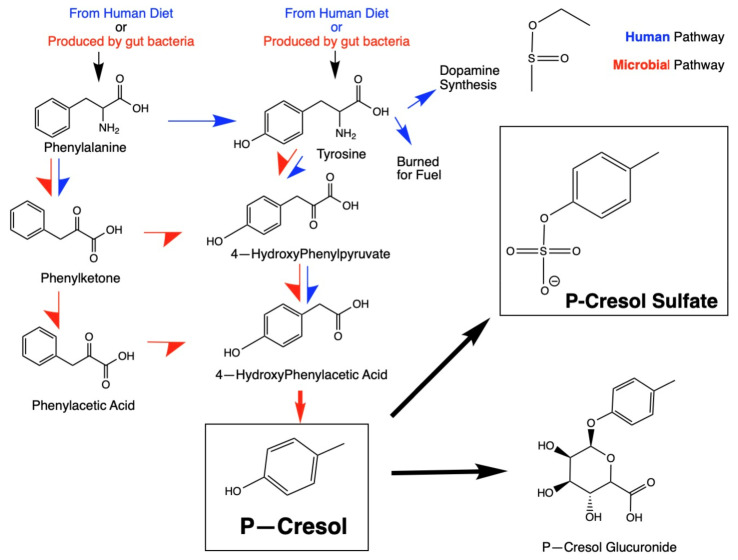
Microbial metabolism and human detoxification metabolic pathways involving p-cresol. Note that microbial metabolic pathways are depicted by red arrows and human pathways are depicted by blue arrows.

**Figure 2 ijms-26-01513-f002:**
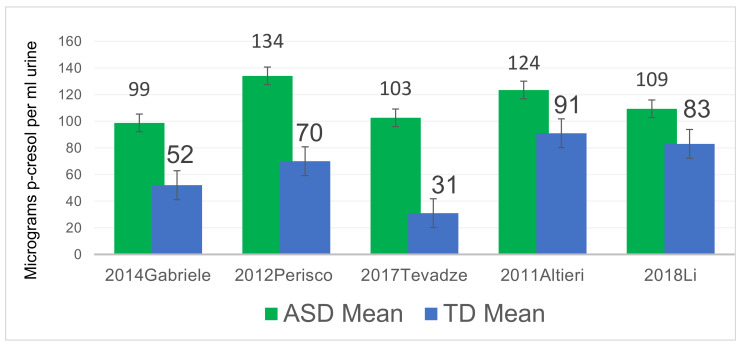
Mean urinary p-cresol concentrations in 5 independent ASD vs. TD urinary metabolomics studies that provided quantitative values in raw urine. Standard error bars are included. 2014 Gabriele [30], 2021 Perisco [37], 2017 Tevzadze [35], 2011 Altieri [29], 2018 Li [36].

**Figure 3 ijms-26-01513-f003:**
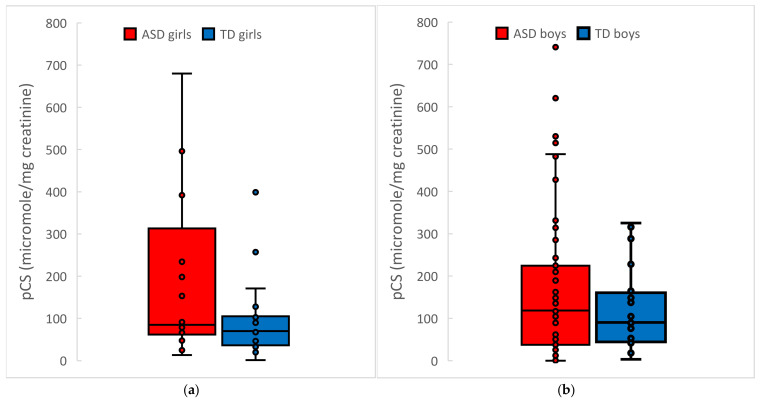
Reanalysis by gender of the urinary pCS levels as measured by Osredkar et al. [38]. The figure shows the median, 1st standard deviation, and 2nd standard deviations. (**a**) *p*-value females = 0.06; (**b**) *p*-value males = 0.005. Not shown are two outliers; one ASD Female pCS = 2161 µmol/mg creatinine and one ASD Male pCS = 1795 µmol/mg creatinine.

**Figure 4 ijms-26-01513-f004:**
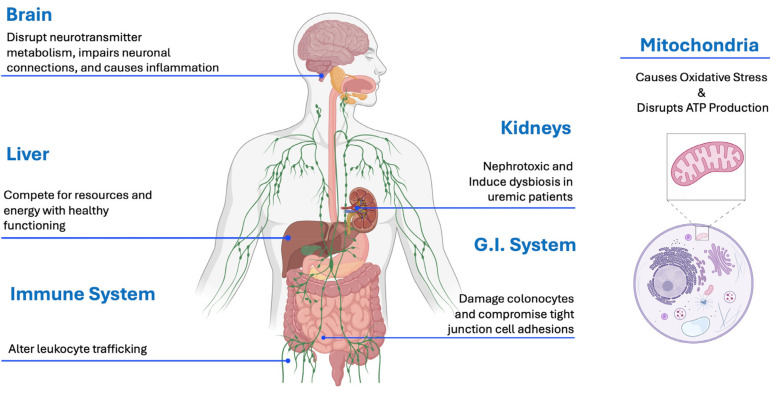
The systemic effects of p-cresol and pCS on multiple organ and cellular functions are discussed in this section.

**Table 1 ijms-26-01513-t001:** Studies of urinary p-cresol and pCS concentrations in ASD participants compared to non-ASD participants. “Significantly elevated” means compared to typically developing controls (*p* < 0.05). “↑” means the mean value in ASD was higher than that of controls, but the statistical significance was not mentioned.

Study	ASD Participants	TD Participants	Country of Origin	Findings
Altieri, 2011 [29]	59	59	Italy	Significantly ↑
Chen, 2013 [41]	156	64	China	Significantly ↑
Daneberga, 2021 [42]	44	**	Latvia	**
Diémé, 2015 [31]	30	32	France	Significantly ↑
Emond, 2013 [32]	26	24	France	Significantly ↑
Gabriele, 2014 [30]	33	33	France	Significantly ↑
Gevi, 2016 [43]	30	30	Italy	↑
Gevi, 2020 [40]	40	40	Italy	Significantly ↑
Li, 2018 [36]	33	44	China	Significantly ↑
Mussap, 2020 [39]	31	26	Italy	Significantly ↑
Noto, 2014 [33]	30	28	France	↑
Osredkar, 2023 [38]	143	48	Slovenia	Significantly ↑
Perisco, 2012 [37]	59	59	Italy	Significantly ↑
Piras, 2022 [34]	13	14	Italy	Significantly ↑
Timperio, 2022 [44]	14	14	Italy	Significantly ↑
Tevzadze, 2017 [35]	14	14	Georgia	Significantly ↑
Zhang, 2020 [45]	39	40	China	↑

** Daneberga 2021 examined microbial community determination and divided the group into “High P-cresol” and “Low P-cresol”.

**Table 2 ijms-26-01513-t002:** Known harmful effects of p-cresol and pCS on the body.

System	Effects
ASD (General)	Causally induces ASD-like behaviors and social deficits in animal experiments
	Urinary levels of P-cresol or pCS sulfate in children with ASD correlated with severity of ASD and ASD-related symptoms
Mitochondria	P-cresol and pCS disrupts mitochondrial function
	Increased reactive oxygen species and oxidative stress
	Depleted glutathione (primary antioxidant)
	Decreased production of ATP (major fuel for body and brain)
Gut	P-cresol is an antibiotic to many gut bacteria species, altering the gut microbiome
	Weakens the intestinal barrier, allowing greater amounts of microbial metabolites to leak into the body
Brain	Competes with neurotransmitter synthesis metabolism as they share a parent compound, phenylalanine
	Impairs dendritic growth and arborization of neurons
	Modulates early brain growth and affects NMDAR and D1 Dopamine receptor development
	Irreversibly inhibits Dopamine-Beta-Hydroxylase, an enzyme vital to catecholamine metabolism
Liver	Competes with bilirubin and other microbial metabolites as it is bound to albumin
	Liver converts p-cresol into p-cresol sulfate, which reduces the sulfate pool for detoxification of other substances, including many neurotransmitters
	Cytochrome P450 enzymes in liver microsomes convert p-cresol into quinone methides, which are highly reactive oxygen species
Kidneys	pCS is highly toxic to the kidneys (nephrotoxin)
	Induces renal tubular cell necrosis in a dose-dependent manner
	Creates oxidative stress and inflammatory responses
	May interfere with filtration efficiency
Immune System	Depresses phagocytic activity of macrophages
	Activates nuclear factor kappa beta signaling
	Reduces Th1 cytokine production
	Alters leukocyte trafficking

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
