# Peer review of "Review of Elevated Para-Cresol in Autism and Possible Impact on Symptoms"

_ijms, 2025, doi:10.3390/ijms26041513_

Round 1

Reviewer 1 Report

Comments and Suggestions for Authors

            This is an interesting and well-structured review describing the generation of p-cresol and its metabolite p-cresol sulfate and their possible influence on autism. The review describes the influence of these factors on the intestine, brain, kidney, liver, immune system and mitochondria in humans and animal models. They also discuss the possible beneficial effects of microbiota transplantation in humans and animals.

            In my opinion, its publication is recommended. However, I think that some "minor points" should be considered before its publication.

Minor points

-On line 22, the opening parenthesis is missing or alternatively the closing parenthesis should be removed.

-On line 33, the definition of GI is missing.

-On line 74, a reference is needed after "colon" [ref].

- On lines 80-81, this last sentence of the paragraph reads weird. Maybe it should be “…hydroxylase, through a series of…”

-It would be interesting to include the enzymes involved in the metabolism and catabolism expressed in figure 1 (page 3).

-On line 89 “uli” should not be with initial capital letter and "and" should not be in italics.

-On line 99 the period should be after the bracket: [29].

- On line 121 there is a dot lacking: “world. Seventeen”

-In Table 1, "TD" has not yet been defined (page 4).

-Generally speaking, when authors cite a reference in the text with the surname of the first author, they should follow it with the number with which it appears in the reference section. For example, on line 138, author citations should be followed by their reference numbers (Altieri et al [30], Gabriele et al [31], [70], Tevzadze et al [55]...).

-On line 138, "Perisco et al" does not appear among the references.

-Likewise, in figure 2, reference numbers should be added where appropriate.

-On line 177 there is an extra open parenthesis.

-On page 8, line 254, the authors indicate "ASD involves increased rates of Parkinson's disease". Since p-Cresol increases dopamine in brain (page 7, lines 207-209) might the author discuss this apparent contradiction?

- On line 301, there should be a comma or “and” after "oxytoca" and "Proteus" should have an initial capital letter.

-On line 312, please check “(OR…

-Lines 338 to 345 need one or more references at the end of the paragraph.

- Throughout the manuscript there are a lot of places where a space should be added. For instance: lines 44 (manifestations[3]), 256 (ASD[13]), 274 (function[58]), 283 ([63].Furthermore), 317 (ASD[70]), 403 (inflammation[50]), 418 (activity[98]).

- Review the complete bibliography because there are numerous errors, for example: In some references the name of the journal is missing or they are incomplete with other elements of the reference. Unify journal titles in abbreviated or complete form.

Author Response

Hello detail oriented reviewer-

Thank you for reviewing my paper.  My answers to your comments are in red below.

This is an interesting and well-structured review describing the generation of p-cresol and its metabolite p-cresol sulfate and their possible influence on autism. The review describes the influence of these factors on the intestine, brain, kidney, liver, immune system and mitochondria in humans and animal models. They also discuss the possible beneficial effects of microbiota transplantation in humans and animals.

Thank you!           

In my opinion, its publication is recommended. However, I think that some "minor points" should be considered before its publication.

Thank you for pointing these out.  I have resolved all of the "Minor Points" you list below, including fixing spacing, references, parathesis, the full version gastrointestinal before abbreviating it to GI, added references where requested, added the customary citation number when discussing a specific study, fixed capitalization issues, and revised the bibliography including adding any missing journals.  Thank you for your attention to detail!

Regarding your question:

-On page 8, line 254, the authors indicate "ASD involves increased rates of Parkinson's disease". Since p-Cresol increases dopamine in brain (page 7, lines 207-209) might the author discuss this apparent contradiction?

 Dopamine exerts cytotoxicity in excess quantities via autoxidation in parts of the brain, and tragically kills the cells responsible for dopamine production located within the the substantia nigra. As PKD progresses, sufferers can’t make dopamine and require synthetic dopamine precursors to control their motor symptoms.  

Citation: Richard Jay Smeyne, Vernice Jackson-Lewis,
The MPTP model of Parkinson's disease,
Molecular Brain Research, Volume 134, Issue 1, 2005, Pages 57-66.  DOI: 10.1016/j.molbrainres.2004.09.017.

Reviewer 2 Report

Comments and Suggestions for Authors
  •  
  • Autism spectrum disorder (ASD) is one of the most frequently occurring neurodevelopmental disorders in childhood and recent reports point to a possible role of the gut–brain axis in ASD. Since ASD diagnosis mainly relies on behavioral evaluations and there are no effective treatments, identifying therapeutical molecular targets for treatment has become increasingly urgent. In the current interesting manuscript the authors, using mostly recent and relevant references. make an extensive review of the literature highlighting the role of p-cresol and its metabolite p-cresol sulfate (pCS) on ASD and other ASD comorbidities; they also discuss the potential therapeutic approach (microbiota transplant therapy) for reducing elevated levels of p-cresol and pCS. Furthermore, it is emphasized the need for continuous efforts and intensive studies in order to investigate a)other microbial-produced metabolites that may also contribute to ASD and co-morbid symptoms, and b) therapeutic approaches. The paper is well organized, well written and fits in the scope of the journal, Figures and tables are very clear and helpful.
  •  

Author Response

  • Autism spectrum disorder (ASD) is one of the most frequently occurring neurodevelopmental disorders in childhood and recent reports point to a possible role of the gut–brain axis in ASD. Since ASD diagnosis mainly relies on behavioral evaluations and there are no effective treatments, identifying therapeutical molecular targets for treatment has become increasingly urgent. In the current interesting manuscript the authors, using mostly recent and relevant references. make an extensive review of the literature highlighting the role of p-cresol and its metabolite p-cresol sulfate (pCS) on ASD and other ASD comorbidities; they also discuss the potential therapeutic approach (microbiota transplant therapy) for reducing elevated levels of p-cresol and pCS. Furthermore, it is emphasized theneed for continuous efforts and intensive studies in order to investigate a)other microbial-produced metabolites that may also contribute to ASD and co-morbid symptoms, and b) therapeutic approaches. The paper is well organized, well written and fits in the scope of the journal, Figures and tables are very clear and helpful.

Thank you for your professional review of my manuscript, and for your kind words!